# Exploration of Light-Controlled Chemical Behavior and Mechanism in a Macrocyclic Copper Complex Catalyst–Acetone–Glucose–Bromate–Sulfuric Acid Oscillation System

**Lin Hu [1],\*, Qujin Cui [1]** 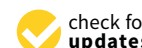 **, Yaqin Zhang [1],\*, Xiaoqin Zhou [1], Yatin Pan [1], Juan Tang [1], Jinqing Li [1], Wenyuan Xu [1] and Hanhong Xu [2]**

[1]   School of Material Science and Engineering, East China JiaoTong University, Nanchang 330013, China; cuiqujin@163.com (Q.C.); zxqlittle@sina.cn (X.Z.); pythonting@sina.com (Y.P.); tj4768745@sina.com (J.T.); lijinqing1993@sina.com (J.L.); xwy1027@sina.com (W.X.)

[2]   China Key Laboratory of Pesticide and Chemical Biology, Ministry of Education, South China Agricultural University, Guangzhou 510642, China; hhxu@scau.edu.cn

\*   Correspondence: hulin21@hotmail.com (L.H.); zyq201812@sina.com (Y.Z.)

**Abstract:** In this paper, the effect of ultraviolet light on the $[CuL](ClO_4)2$–glucose (Glu)–acetone (Act)–sodium bromate ($NaBrO_3$)–sulfuric acid ($H_2SO_4$) oscillation system was studied. The reaction mechanism and Oregonator model were established to verify the mechanism. Comparison of the bromide ion electrode–platinum electrode correlation diagrams with and without ultraviolet light reveals a nontracking phenomenon in the bromide ion electrode–platinum electrode correlation diagram under illumination, indicating that the illumination will affect the changes in the bromide ion concentration in the system. During the process, as UV intensity increases, the concentration of bromide ions in the system increases, and bromide ions can inhibit chemical oscillations, resulting in a decrease in the amplitude of chemical oscillations, further verifying that the proposed mechanism is reasonable.

**Keywords:** B-Z oscillations; univariate method; ultraviolet light; oregonator model

## 1. Introduction

Oscillatory chemical reactions [1,2], which are very sensitive, are disturbed by various small interfering factors, making them useful as an advanced analytic technique for trace substances. These systems generally include the Belousov–Zhabotinsky(BZ) [3–5], Bray–Liebhafsky(BL) [6], and Briggs-Rauscher(BR) [7] oscillation systems. In particular, BZ reactions have been extensively studied to elucidate their mechanisms and to model the wider range of excitable media.

In recent years, it was proposed that light affects the BZ oscillation reaction [8,9], and experiments that combined light and BZ oscillation reactions have proved the photosensitive properties of this reaction, making it a hot topic for the scientific community. Photochemical reactions [10] are one of the most important reaction processes in the life cycle. For example, both photosynthesis [11] and visual reactions are photochemical reactions. Therefore, the study of the effect of light interference on the BZ oscillation system can provide insight into many poorly understood phenomena in biological systems.

Many hypothetical mechanisms of the photosensitive BZ oscillation reaction [12] have been proposed and discussed, gradually deepening the understanding of the reaction. Additionally, the BZ oscillating photochemical reaction catalyzed by $Ru(bpy)_3^{2+}$ [13,14] has drawn significant attention; this reaction is highly sensitive to disturbance by light and can transition from photoinhibition to

photoexcitation, so the response to light can be used to explore the mechanism of this reaction. The perturbation of the BZ oscillation system, which is caused by light irradiation, changes how both light intensity and light source distribution affect the original amplitude and period; within a certain range, the light intensity increases, and the oscillation reaction gradually decreases and then terminates. It can be seen that the interference effect of the light during oscillation-limits oscillatory behavior. A photosensitive BZ oscillatory reaction mechanism is a slight modification of the typical Oregonator(FKN) mechanism [15] that is described by the following three formulas:

$$A: \quad BrO_3^- + 5Br^- + 6H^+ \rightarrow 3Br_2 + 3H_2O$$
$$B: \quad BrO_3^- + HBrO_2 + 2C_{red} + 3H^+ \rightarrow 2HBrO_2 + 2C_{oxi} + H_2O$$
$$C: \quad 2C_{oxi} + MA + BrMA \rightarrow 2C_{red} + fBr^- + other\ things$$

Here, $C_x$ is a catalyst (where x represents its state, oxi means oxidation, and red means reduction). According to the FKN mechanism, in the absence of light, BZ oscillations exhibit stable fluctuations through the above three reactions. Ru(bpy) (II) is used as a catalyst for sensitization, and the sensitometric properties of B-Z chemical oscillations are strongly related to the characteristics of pyridinium ruthenium (II)-3,4. Experimentally, it was found that the optical interference of the B-Z oscillation effect may be mainly due to the following reaction:

$$Ru(bpy)_3^{2+} + h\nu \rightarrow Ru(bpy)_3^{2+\bullet}$$

The Ru(bpy)$_3{}^{2+\bullet}$ produced by this reaction is strongly reducing, reacts with various substances throughout the process and gives rise to the light interference reaction. Ru(bpy)$_3{}^{2+\bullet}$ reacts with other substances as follows:

$$a: \quad 6Ru(bpy)_3^{2+\bullet} + BrO_3^- + 6H^+ \rightarrow Ru(bpy)_3^{2+} + 3H_2O + Br^-$$
$$b: \quad 4Ru(bpy)_3^{2+\bullet} + HBrO_2 + 3H^+ \rightarrow 4Ru(bpy)_3^{2+} + 2H_2O + Br^-$$
$$c: \quad Ru(bpy)_3^{2+\bullet} + BrMA \rightarrow Ru(bpy)_3^{2+} + Br^- + other\ things$$
$$d: \quad Ru(bpy)_3^{2+\bullet} + BrMA \rightarrow Ru(bpy)_3^+ + H^+ + CO_2 + Br^-$$
$$e: \quad Ru(bpy)_3^{2+\bullet} + BrMA \rightarrow Ru(bpy)_3^{3+} + Br^- + other\ things$$
$$\quad Ru(bpy)_3^{2+\bullet} + BrMA \rightarrow Ru(bpy)_3^{2+} + Br^- + other\ things$$
$$f: \quad Br_2 + BrMA \rightarrow Br_2MA + H^+ + Br^-$$
$$\quad Ru(bpy)_3^{2+\bullet} + Br_2MA \rightarrow Ru(bpy)_3^{3+} + Br^- + MA$$

For processes a–f, although Ru(bpy)$_3{}^{2+\bullet}$ reacts with substances such as $BrO_3{}^-$, $HBrO_2$, BrMA, and Br$_2$MA, the common feature of all the reactions is that Br$^-$ is present in the final product. It is observed that the position of Br$^-$ in the entire reaction cannot be ignored, and Br$^-$ hinders the oscillating reaction, which is the key factor that hinders the oscillating phenomenon after light perturbs the oscillating reaction.

Chemical oscillations can reflect the characteristics of non-linear phenomena and simulate biological behaviors. Based on a previous report, to expand the family of photochemical oscillatory systems, we study the $H_2SO_4$-NaBrO$_3$-Acetone-Glucose B-Z oscillation system catalyzed by [CuL](ClO$_4$)$_2$ [16,17] and predict that the system exhibits photosensitivity. Compared to some chemical oscillations (systems catalyzed by $Mn^{2+}$, $Ce^{4+}$), the oscillation reaction of [CuL](ClO$_4$)$_2$ is particularly relevant, because the macrocyclic complex has a structure similar to an enzyme, where ligand L is 5,7,7,12,14,14-hexemethyl-1,4,8,11-tetraazacyclo tetradeca_4,11-diene. Therefore, the study of [CuL](ClO$_4$)$_2$ participating in the redox reaction is particularly valuable for exploring the chemical changes in the process. In this experiment, a mixture of glucose and acetone was chosen. Glucose is an indispensable substance in biological processes. Thus, this chemical oscillation system is related to a biological system.



In the present work, we explored the photochemical properties of the above system. The system has an ultra-sensitive response to ultraviolet light, and the amplitude changes significantly but the frequency remained unaffected under ultraviolet light with a wavelength of 245 nm and light intensity of 450–1300 Lux. At a constant light intensity (1250 Lux), the concentrations of $NaBrO_3$, glucose, acetone, and $[CuL](ClO_4)_2$ are favored. The photochemical phenomena of the system were studied by using platinum electrode and a bromide ion selective electrode as the working electrodes in the same oscillating solution.

Most importantly, we mathematically simulated of the behavior of the chemical oscillations based on the photosensitive reaction equation. We analyzed and simplified the chemical reaction mechanism, and established a model of chemical oscillation, and a series of chemical oscillation processes after mathematical processing. This set of equations provides the mathematical model of chemical oscillations. By selecting the appropriate mathematical software (i.e., Mathematics) to solve the equations, the solutions of the differential equations are expressed as images to reflect the oscillations [18].

## 2. Results and Discussion

### 2.1. Optimization of the Reactant Concentration

The concentrations of sodium bromate, glucose, acetone and $[CuL](ClO_4)_2$ in the oscillating system were optimized under constant light exposure (1250 Lux). We assume that $A_0$ is the amplitude at the initial concentration, $A_x$ is the amplitude at the other concentration, and $\Delta A$ is the difference between them, i.e., $\Delta A = A_0 - A_x$.

The experiments were conducted under a constant light intensity (1250 Lux) and with the initial concentrations $[H_2SO_4]_0 = 1.05$ M, $[Acetone]_0 = 0.65$ M, $[Glucose]_0 = 0.06$ M, and $[[CuL](ClO_4)_2]_0 = 0.05$ M. A single variable method was used to optimize the concentrations of $H_2SO_4$, acetone, glucose and $[CuL](ClO_4)_2$.

The initial concentrations of other substances did not change, the effect of the optical interference oscillating system was studied for initial concentrations of sodium bromate ($[Sodium\ Bromate]_0$) from 0.04 M to 0.14 M, and the relationship between the reaction of the optical interference oscillating system and the concentration of sodium bromate was determined. The results show that as concentration increases, $\Delta A$ first increases and then decreases before reaching a maximum value. (Figure 1A). This method was also used to study the effects of the changes in the concentrations of acetone (0.45 M to 1.25 M), glucose (0.04 M to 0.12 M) and $[CuL](ClO_4)_2$ (0.03 M to 0.11 M), and the optimal concentrations were determined (Figure 1B–D, respectively).

The experimental results show that, as concentration increases, $\Delta A$ first increases, and then when the concentration reaches a critical value, $\Delta A$ decreases with further increase in concentration. The optimal concentrations of the substances are: $[H_2SO_4]_0 = 1.05$ M, $[Acetone]_0 = 0.65$ M, $[Glucose]_0 = 0.06$ M, $[Sodium\ Bromate]_0 = 0.08$ M, and $[[CuL](ClO_4)_2]_0 = 0.05$ M.

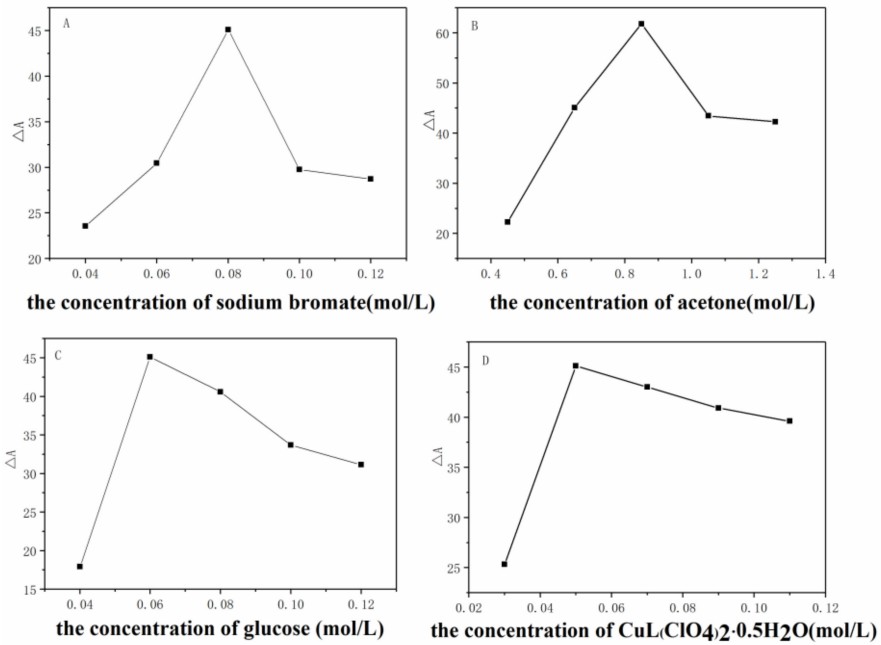

**Figure 1.** The effect of light while varying the concentration of: (**A**) sodium bromate; (**B**) acetone; (**C**) glucose; (**D**) CuL(ClO₄)₂. Common conditions: $[H_2SO_4]_0 = 1.05$ M; ultraviolet light applied at 245 nm with an intensity of 1250 Lux.

### 2.2. Analysis of B-Z Oscillation System Behavior under UV Light

The initial concentrations of the components in the oscillating system are: $[\text{Sulfuric acid}]_0 = 1.05$ M, $[\text{Acetone}]_0 = 0.65$ M, $[\text{Glucose}]_0 = 0.06$ M, $[\text{Sodium Bromate}]_0 = 0.08$ M, and $[[\text{CuL}](\text{ClO}_4)_2]_0 = 0.05$ M. The platinum electrode and the Br-electrode are used as the working electrodes, and oscillation patterns with different effects were obtained. At typical oscillation plot in the absence of ultraviolet radiation is shown in Figure 2.

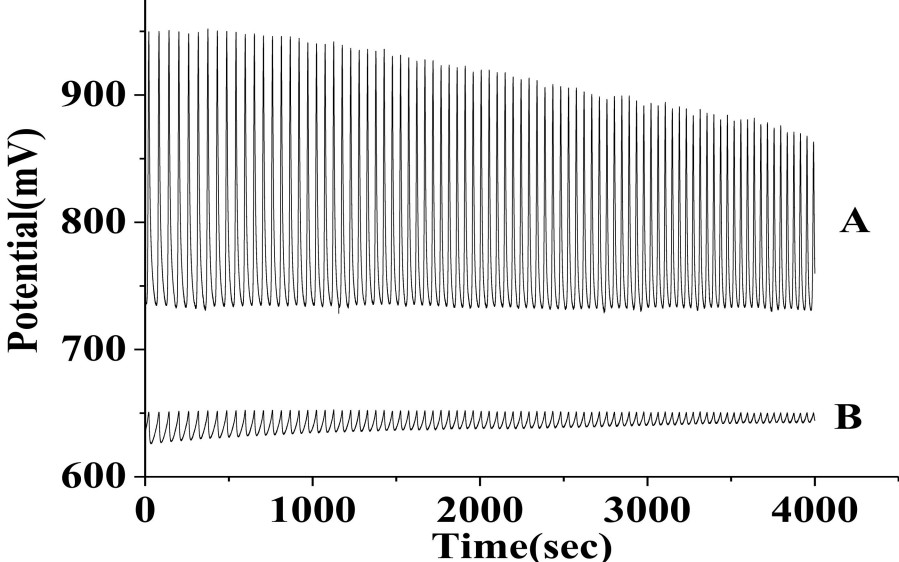

**Figure 2.** Typical potential oscillation profiles in the absence of light (voltage-time): (**A**) platinum electrode, (**B**) bromide ion selective electrode. Common conditions: $[H_2SO_4]_0 = 1.05$ mol/L; $[\text{Acetone}]_0 = 0.65$ mol/L; $[\text{Glucose}]_0 = 0.06$ mol/L; $[\text{Sodium bromate}]_0 = 0.08$ mol/L; $[[\text{Cu}](\text{ClO}_4)_2]_0 = 0.05$ mol/L.

After the oscillation was stable for 1000 s, we turned on the UV lamp (UV wavelength of 245 nm, light intensity of 1250 Lux) to irradiate the oscillating solution and studied the effects of ultraviolet

light on the oscillating system. We found that the amplitude of oscillation is significantly reduced by ultraviolet light, but the frequency is not substantially affected. When the UV lamp was turned off, the oscillation returns to the typical oscillation state (Figure 3).

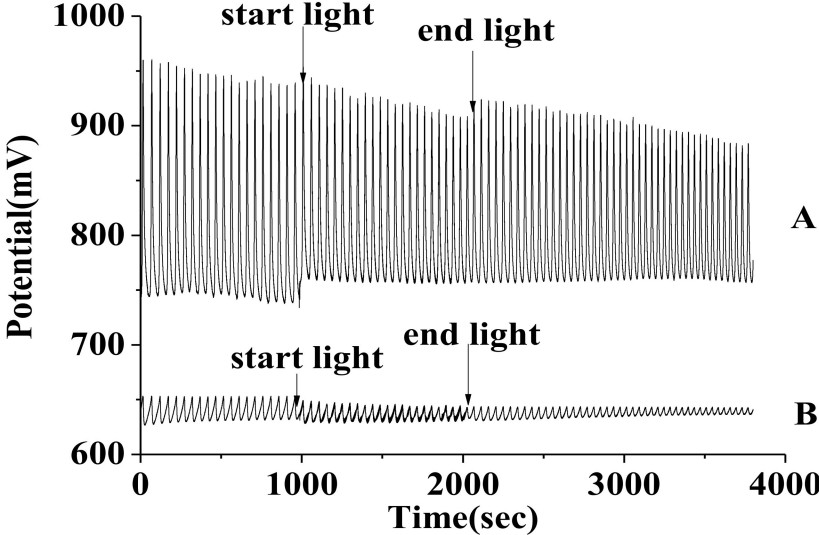

**Figure 3.** Typical potential oscillation profiles in the presence of light: (**A**) platinum electrode, (**B**) bromide ion selective electrode. Common conditions: $[H_2SO_4]_0 = 1.05$ mol/L; $[Acetone]_0 = 0.65$ mol/L; $[Glucose]_0 = 0.06$ mol/L; $[Sodium\ bromate]_0 = 0.08$ mol/L; $[[Cu](ClO_4)_2]_0 = 0.05$ mol/L. Ultraviolet light applied at 245 nm with an intensity of 1250 Lux.

According to the Nernst equation given by $E = E^{\ominus} + \frac{RT}{nF}\lg\frac{[Br]_O}{[Br^-]}$, the absolute voltage of oscillation increases with bromide ion concentration (Figure 2A), and it was found that the absolute voltage of oscillation rose when light irradiation started, showing that applied light can perturb oscillation. When the light was turned on, the absolute voltage increased but the change in voltage (i.e., amplitude) decreased, implying that the rage of change of bromide ion concentration was decreasing, which indicated that the bromide ion concentration in the system was related to the change in the amplitude.

### 2.3. Analysis of Different Light Intensity Interference Oscillation Systems

Using the optimized concentrations ($[H_2SO_4]_0 = 1.05$ M, $[acetone]_0 = 0.65$ M, $[glucose]_0 = 0.06$ M, $[sodium\ bromate]_0 = 0.08$ M, $[[CuL](ClO_4)_2]_0 = 0.05$ M), we studied the effects of different light intensities on the BZ oscillation system. Here, A is the actual amplitude oscillation under UV illumination, and A0 is the ideal amplitude that is assumed to be simulated without UV illumination. ΔA is the difference between the actual amplitude under illumination and the ideal amplitude, i.e., $\Delta A = A_0 - A$, and $\Delta A\% = (A_0 - A)/A$. Positive ΔA and ΔA% indicate the inhibitory effect of light on the oscillating reaction.

Then, we studied the disturbance of the oscillation behavior at different light intensities. The change in amplitude of oscillation (ΔA) increased with increasing light intensity from 450 Lux to 1250 Lux (Figure 4), indicating that increased light intensity can increase the inhibition of oscillation. The effect of light intensity found in this work on the amplitude of the oscillating system is described by the following linear expression:

$$\Delta A\% = 0.0419 \times Light\ intensity + 54.00$$

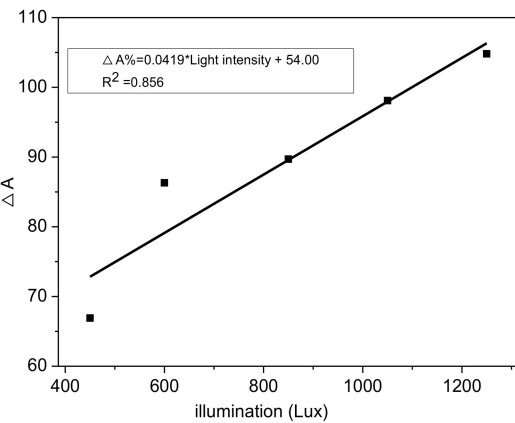

**Figure 4.** Change in amplitude against light intensity. Common conditions: $[H_2SO_4]_0 = 1.05$ M; $[Acetone]_0 = 0.65$ M; $[Glucose]_0 = 0.06$ M; $[Sodium\ bromate]_0 = 0.08$ M; $[CuL](ClO_4)_2]_0 = 0.05$ M; ultraviolet light applied at 245 nm.

In addition, we created a correlation diagram for the potential of the platinum electrode and the Br-selective electrode at each same time point, and observed the effect of light on the overall oscillation response trajectory (Figure 5). As seen in the figure, when UV light irradiates the oscillating system, there are signs of losing track (Figure 5).

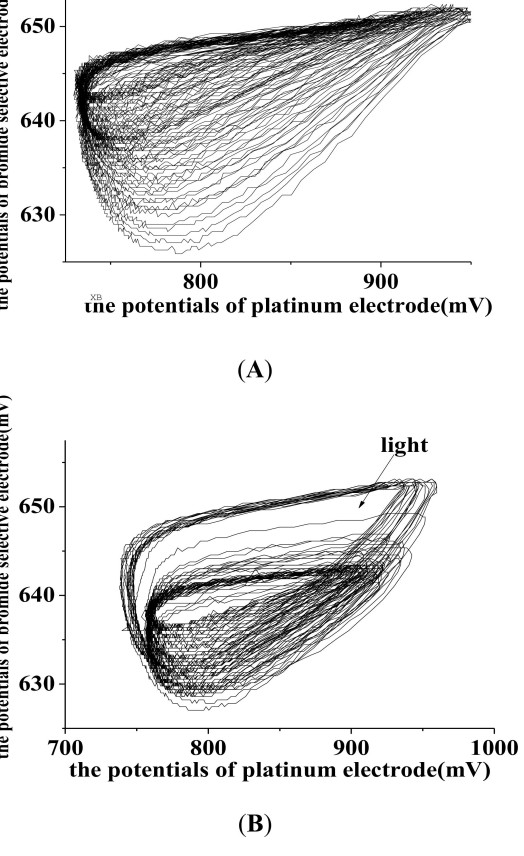

**Figure 5.** (**A**) Trajectory of the potential of the platinum electrode against the potential of the bromide selective electrode. (**B**) Effect of light on the trajectory of the potential of the platinum electrode against the potential of the bromide selective electrode on oscillation; ultraviolet light applied at 245 nm with an intensity of 1250 Lux. Common conditions: $[H_2SO_4]_0 = 1.05$ M; $[Acetone]_0 = 0.65$ M; $[Glucose]_0 = 0.06$ M; $[Sodium\ bromate]_0 = 0.08$ M; $CuL[(ClO_4)_2]_0 = 0.05$ M.

This effect may be explained as follows: under light irradiation, the bromide ion concentration in the oscillating system changes rapidly, increasing the bromide ion potential. Large changes in bromide ion potential lead to large changes in the correlation between the platinum electrode potential and the bromine electrode potential, generating a clear gap in the figure.

## 3. Proposed Mechanism

The mechanism for the Cu(II)—H2SO4—Acetone—Glucose—NaBrO3 system in the absence of light was proposed to be expressed by the following Equations (1)–(9).

$$BrO_3^- + Br^- + 2H^+ \rightleftharpoons HBrO + HBrO_2 \tag{1}$$

$$HBrO_2 + Br^- + H^+ \rightleftharpoons 2HBrO \tag{2}$$

$$HBrO + Br^- + H^+ \rightleftharpoons Br_2 + H_2O \tag{3}$$

$$BrO_3^- + HBrO_2 + H^+ \rightleftharpoons 2BrO_2 + H_2O \tag{4}$$

$$Br_2 + CH_2OH(CHOH)_3CHOHCHO \\ \rightarrow Br^- + H^+ + CH_2OH(CHOH)_3CBrOHCHO \tag{5}$$

$$Br_2 + CH_3COCH_3 \rightarrow Br^- + H^+ + CH_3COCH_2Br \tag{6}$$

$$BrO_2 + [CuL]^{2+} + H^+ \rightarrow [CuL]^{3+} + HBrO_2 \tag{7}$$

$$CH_2OH(CHOH)_3CBrOHCHO + 2[CuL]^{3+} + HBrO + H_2O \\ \rightarrow CH_2OH(CHOH)_3COOH + CO_2 + 4H^+ + 2[CuL]^{2+} \\ + 2Br^- \tag{8}$$

$$CH_2OH(CHOH)_3CBrOHCHO + [CuL]^{3+} + H_2O \\ \rightarrow [CuL]^{2+} + Br^- + CO_2 + H^+ \tag{9}$$

The reaction process of (8) and (9) are quite complex, and the products are diverse, which can be ultimately simplified into equation C in Section 4. When light irradiates the oscillating system, the possible mechanism of the light irradiation effect may be described by:

$$Br_2 \overset{light}{\rightarrow} 2Br\bullet \tag{1a}$$

$$Br\bullet + [CuL]^{2+} + H^+ \rightarrow Br^- + H^+ + [CuL]^{3+} \tag{2a}$$

Under UV light irradiation, $Br_2$ in reaction (3) will generate bromine radicals that are highly unstable and strongly oxidizing; these radicals oxidize $[CuL]^{2+}$ to $[CuL]^{3+}$, and at the same time are reduced to bromide ions. Overall, the concentration of oxidizing substances ($[CuL]^{3+}$, $BrO_3^-$, Br, etc.) increases and the concentration of reducing substances ($[CuL]^{2+}$, $Br^-$) decreases under irradiation. In the special equation, the absolute voltage of the system rises, in agreement with experimental observations.

## 4. Oregonator Model

The FKN mechanism discussed above and the Oregonator model [19] based on the FKN mechanism are well-known. The model involves three processes (including five chemical reactions) and can be expressed by the following reaction equations:

$$A: \quad 2H^+ + BrO_3^- + Br^- \xrightarrow{k_1} HBrO_2 + HBrO$$
$$H^+ + HBrO_2 + Br^- \xrightarrow{k_2} 2HBrO$$
$$B: \quad BrO_3^- + 2[CuL]^{2+} + HBrO + H^+ \xrightarrow{k_3} 2HBrO_2 + 2[CuL]^{3+}$$
$$2HBrO \xrightarrow{k_4} BrO_3^- + HBrO + H^+$$
$$C: \quad C_{red} + [CuL]^{3+} + mHBrO \xrightarrow{k_5} nBr^- + [CuL]^{2+}$$

The $k_x$ in the above equations are the rate constants, where x represents the first few steps and n is the number of Br(II) produced by Cu(II) throughout the oscillation. Here, we use symbols for the above substances: $BrO_3^-$ is denoted by A, HOBr is denoted by P, $HBrO_2$ is denoted by X, $Br^-$ is denoted by Y, and $Cu^{2+}$ is denoted by Z; the ordinary differential equations are written using these symbols. In Equation C, [CuL]3+ and $Br^-$ are not deterministic quantitative relations. N represents the number of Br- that can be catalyzed by $[CuL]3^+$. The equations can be expressed as follows:

$$①dx/dt = k_1 AY - k_2 XY + k_3 AX + 2k_4 X^2$$
$$②dy/dt = -k_1 AY - k_2 XY + nk_5 Z$$
$$③dz/dt = 2k_3 AX - k_5 Z$$

In the above ordinary differential equations, the expressions are related to the dimension of the units (time-concentration relationship). When the simulation is performed, the equation must be processed dimensionlessly (i.e., converted to a unit with a dimension of 1). Using x, y, z, τ, ε, p, and q to represent various substances, we obtain:

$$x = k_2 X/(k_5 A), \quad y = k_2/(k_3 A)Y, \quad z = k_2 k_5 Z/\left(2k_1 k_3 A^2\right),$$
$$e = k_1/k_3, \quad p = k_1 A/k_5, \quad q = 2k_1 k_4/(k_2 k_3)$$

After processing, a unit of 1 is obtained, and the equations are given by:

$$④\varepsilon(dx/dt) = x + y - xy - qx^2, \quad x(0) = a$$
$$⑤dy/dt = 2nz - y - xy, \quad y(0) = b$$
$$⑥p(dz/dt) = x - z, \quad z(0) = c$$

The mathematical calculation software Mathematica 5.0 [20] is suitable for simulating the entire oscillation process. In the computational simulations, the concentration of each substance in the oscillating reaction system at any time point is used as the simulation standard, and the component substances are calculated by the software. Using the software, the numerical solution of the concentration is then converted into a graphical solution to represent the entire oscillation process in a visual manner.

Based on the oscillation experiment, a simple model diagram was created based on the actual oscillation diagram using Mathematica 5.0, and the proposed mechanism was analyzed by analyzing the model diagram.

The observed oscillation B-Z in the absence of light is shown in Figure 6A, and the observed oscillation pattern (potential-time) of the system under light irradiation is shown in Figure 6B.

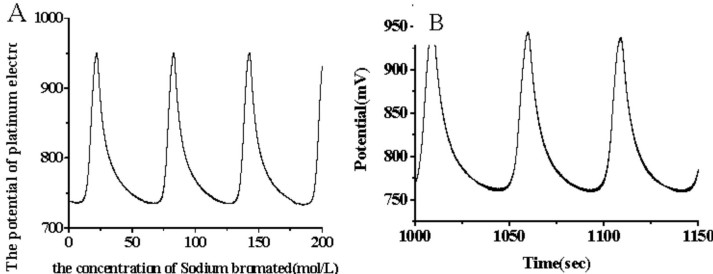

**Figure 6.** Observed oscillation pattern (potential-time); (**A**) in the absent of light; (**B**) under light irradiation. Common system: $[CuL](ClO_4)_2$_$H_2SO_4$_Acetone_Glucose_Sodium bromate.

First, we simulated the basic chemical oscillation model without light, as shown in Figure 7.

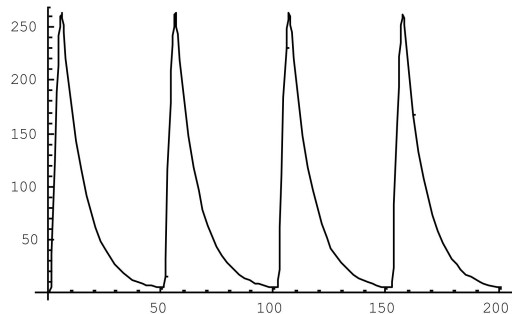

**Figure 7.** Diagram obtained via simulation.

The parameters of the simulation graph are: $k_1 = 2\,M^{-1}s^{-1}$, $k_2 = 2 \times 108\,M^{-1}s^{-1}$, $k_3 = 0.25\,M^{-1}s^{-1}$, $k_4 = 1.25 \times 106\,M^{-1}s^{-1}$, $k_5 = 0.3\,M^{-1}s^{-1}$, $\varepsilon = 8$, $p = 0.1$, $q = 0.001$, $n = 0.55$, $A = 0.015\,mol/L$.

Based on the examination of the chemical oscillation diagram obtained by the illumination chemical oscillation system, it was concluded that the amplitude of the oscillating reaction will be reduced but the period has no significant change. Therefore, by changing the parameters of the above model, we can obtain a model diagram for which the period is unchanged, but the amplitude is reduced (Figure 8).

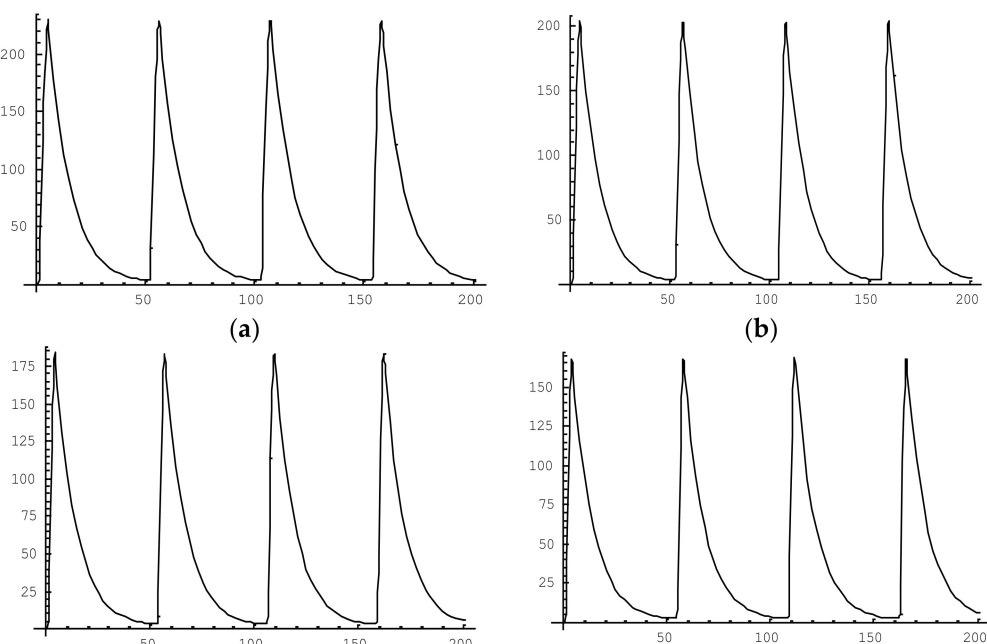

**(a)**

**(b)**

**Figure 8.** *Cont.*

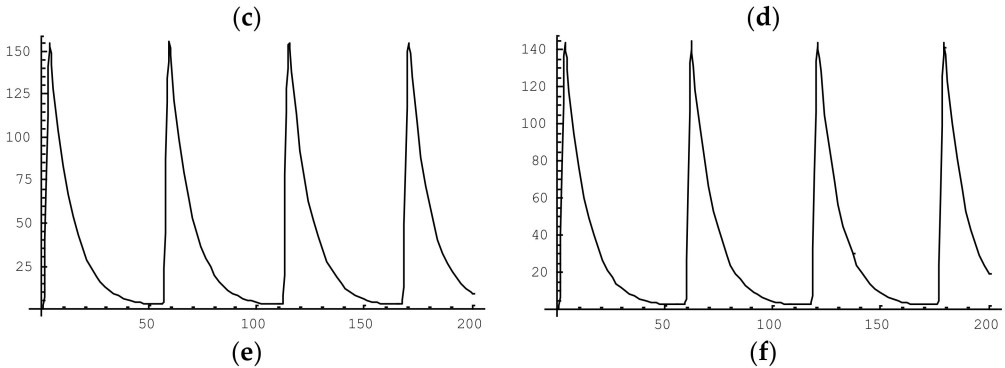

**Figure 8.** Diagram of the simulation under light irradiation; the six graphs represent oscillation simulation diagrams for different light intensity.

The parameters used to obtain Figure 8a are as follows: $k_1 = 2\ M^{-1}s^{-1}$, $k_2 = 3{,}000{,}000\ M^{-1}s^{-1}$, $k_3 = 10{,}000\ M^{-1}s^{-1}$, $k_4 = 6{,}750{,}000\ M^{-1}s^{-1}$, $k_5 = 0.0027\ M^{-1}s^{-1}$, $\varepsilon = 0.0002$, $p = 11.1$, $q = 0.0009$, $n = 0.65$, $A = 0.015\ mol/L$.

The parameters used to obtain Figure 8b are as follows: $k_1 = 2\ M^{-1}s^{-1}$, $k_2 = 3{,}000{,}000\ M^{-1}s^{-1}$, $k_3 = 10{,}000\ M^{-1}s^{-1}$, $k_4 = 6{,}750{,}000\ M^{-1}s^{-1}$, $k_5 = 0.0027\ M^{-1}s^{-1}$, $\varepsilon = 0.0002$, $p = 11.1$, $q = 0.0009$, $n = 0.75$, $A = 0.015\ mol/L$.

The parameters used to obtain Figure 8c are as follows: $k_1 = 2\ M^{-1}s^{-1}$, $k_2 = 3{,}000{,}000\ M^{-1}s^{-1}$, $k_3 = 10{,}000\ M^{-1}s^{-1}$, $k_4 = 6{,}750{,}000\ M^{-1}s^{-1}$, $k_5 = 0.0027\ M^{-1}s^{-1}$, $\varepsilon = 0.0002$, $p = 11.1$, $q = 0.0009$, $n = 0.85$, $A = 0.015\ mol/L$.

The parameters used to obtain Figure 8d are as follows: $k_1 = 2\ M^{-1}s^{-1}$, $k_2 = 3{,}000{,}000\ M^{-1}s^{-1}$, $k_3 = 10{,}000\ M^{-1}s^{-1}$, $k_4 = 6{,}750{,}000\ M^{-1}s^{-1}$, $k_5 = 0.0027\ M^{-1}s^{-1}$, $\varepsilon = 0.0002$, $p = 11.1$, $q = 0.0009$, $n = 0.95$, $A = 0.015\ mol/L$.

The parameters used to obtain Figure 8e are as follows: $k_1 = 2\ M^{-1}s^{-1}$, $k_2 = 3{,}000{,}000\ M^{-1}s^{-1}$, $k_3 = 10{,}000\ M^{-1}s^{-1}$, $k_4 = 6{,}750{,}000\ M^{-1}s^{-1}$, $k_5 = 0.0027\ M^{-1}s^{-1}$, $\varepsilon = 0.0002$, $p = 11.1$, $q = 0.0009$, $n = 0.105$, $A = 0.015\ mol/L$.

The parameters used to obtain Figure 8f are as follows: $k_1 = 2\ M^{-1}s^{-1}$, $k_2 = 3{,}000{,}000\ M^{-1}s^{-1}$, $k_3 = 10{,}000\ M^{-1}s^{-1}$, $k_4 = 6{,}750{,}000\ M^{-1}s^{-1}$, $k_5 = 0.0027\ M^{-1}s^{-1}$, $\varepsilon = 0.0002$, $p = 11.1$, $q = 0.0009$, $n = 0.115$, $A = 0.015\ mol/L$.

In the above parameters, n represents the light intensity. It can be seen from subfigures (a) to (f) that the amplitude decreases with the increase of n and the oscillation period remains unchanged.

An examination of the diagram of the observed chemical oscillation (Figure 9) shows that, in the absence of light, the amplitude of the chemical oscillation is approximately 250 mV, which corresponds to the amplitude of the oscillation pattern simulated when n = 0.55, so that n is set to 0.55. For a graph of simulated oscillation in the absence of light, as n increases, the amplitude of the simulated oscillation decreases, with a good linear relationship between the rate of change of the amplitude and n (Figure 9A). According to Figure 9B, n of the simulation system increases with the increase in the light intensity, and there is a good linear relationship between n and light intensity. Furthermore, we can obtain the relationship between the light intensity and the rate of change of the amplitude of the simulated oscillation graph. Comparing the rate of change of the amplitude obtained experimentally with the rate of change obtained via simulation shows that the simulation system is similar to the experimental system, indicating that the use of simulation is valid. The system illustrates the reaction mechanism of the experiment.

The simulations show that other parameters in the light simulation map do not change. This result is because n changes, and this change gives rise to a change in Br-, in turn increasing n and giving rise to an increase in Br- concentration, which hinders the oscillation reaction. The effect of the oscillating reaction results in a decrease in the amplitude as a result of an increase in the bromide ion concentration.

Of course, more complex mechanisms still need to be explored.

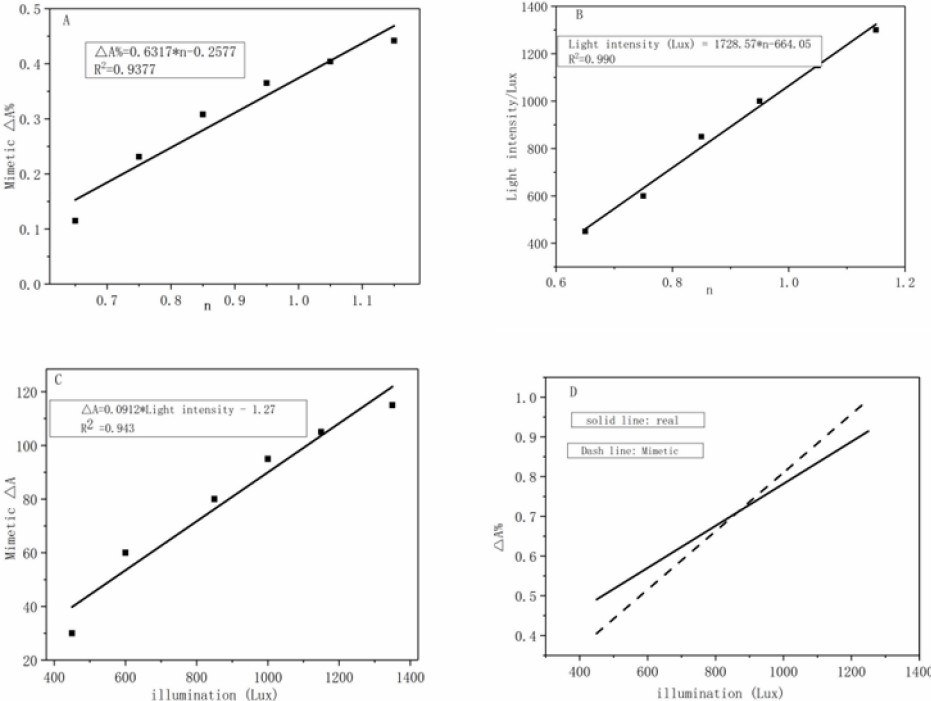

**Figure 9.** (**A**) Linear relationship between n and the rate of change in the amplitude of the simulated oscillation; (**B**) linear relationship between n and the actual intensity; (**C**) rate of change of the amplitude of the light intensity versus the simulated oscillation; (**D**) linear relationship between actual light intensity and the rate of change of the amplitude of the actual oscillation pattern and the simulated oscillation pattern.

## 5. Materials and Methods

### 5.1. Materials and Instruments

An electronic analytical balance (JM-B2006) was purchased from Zhuji Chaozehengqi Equipment Co., Ltd., Zhuji, China. 217 double salt bridge calomel electrodes, platinum electrodes and bromide ion selective electrodes were purchased from Wuhan Gaoss Union Technology Co., Ltd., Wuhan, China. An electrochemical workstation (CHI660D) was purchased from Shanghai Chenhua Instrument Co., Ltd., shanghai, China. An ultraviolet light was purchased from Hangzhou Qiwei Instrument Co., Ltd., hangzhou, China. A digital display temperature control magnetic stirring instrument was purchased from Xiangtan Zhongda Equipment Factory Co., Ltd., Xiangtan, China. The $H_2SO_4$(AR), Acetone(AR), [CuL](ClO4)2(AR), Glucose(AR) and sodium bromate (AR) were purchased from Tianjin Damao Chemical Reagent Factory Co., Ltd., Tianjin, China.

### 5.2. Methods

A 50-mL beaker, which was used as a container for the oscillation reaction, was held at a temperature of $25 \pm 0.5$ °C. A 217 double salt bridge calomel electrode and a platinum electrode (or bromide ion selective electrode) were used as the reference electrode and the working electrode, respectively. $H_2SO_4$ (20.00 mL, 1.05 M), acetone (5.00 mL, 0.65 M), [CuL](ClO$_4$)$_2$ (5.00 mL, 0.05 M), glucose (5.00 mL, 0.06 M), and sodium bromate (5.00 mL, 0.08 M) were mixed in the 50-mL beaker, and the total mixed solution is an oscillating system. To stabilize the chemical oscillating system, the mixed solution was stirred with a digital temperature-controlled magnetic stirrer. The water bath was controlled at a temperature of $25 \pm 0.5$ °C, and the stirring rate was 540 rpm. The reaction was performed by monitoring the potential changes of the platinum electrode and the Br selective electrode.

The run time and sample interval parameters of the electrochemical workstation software were set to 3500 s and 0.05, and the oscillatory reaction was recorded; as shown by the general B-Z oscillation diagrams, a period of steady oscillation was quickly reached.

To determine the effect of light on the BZ chemical oscillations, a continuously variable light UV lamp was used as a light source to irradiate the oscillating system when oscillation was relatively stable (usually approximately 1000 s). The distance between the lamp and the beaker was 5 cm. After a period of time, the effect of light on the BZ chemical oscillation system was observed, and the corresponding parameters (i.e., amplitude and period) were recorded. Finally, under these optimal conditions, the functional relationship between different light intensities and the parameters of the oscillating system was examined.

## 6. Conclusions

For the first time, the effect of light on the $[CuL](ClO_4)_2$–glucose (Glu)–Acetone ($NaBrO_3$)–sulfuric acid ($H_2SO_4$) system was studied; in this system, $[CuL](ClO_4)_2$ has a catalytic effect, and glucose and acetone are mixed substrates. To better understand this new type of chemical oscillation system and the effect of light on this system, we conducted experiments to optimize the effect of light on the system. Under optimal concentration conditions, the relationship among oscillation amplitude, oscillation period, and light intensity were studied, and experimental analysis revealed the linear relationship between the amplitude change and the light intensity. In addition, the correlation diagram of the voltage of the oscillating system simultaneously measured by a platinum electrode and a bromide ion selective electrode was obtained, and the possible mechanism of the oscillating system was proposed.

Based on the experiments, using Mathematica 5.0, we obtained a simple model diagram using the actual oscillation diagram, and analyzed the proposed mechanism using the model diagram. The other parameters in the light simulation map do not change, because n has changed, and the change in n gives rise to the change of $Br^-$ concentration, in turn increasing n, which then increases the $Br^-$ concentration; the bromide ion hinders the oscillating reaction. The subsequent increase in the amount of bromide ions results in a decrease in the amplitude. More complicated mechanisms should be explored in future work.

**Author Contributions:** Conceptualization, L.H. and H.X.; Data curation, Q.C., Y.Z., X.Z. and J.L.; Formal analysis, Y.Z. and J.T.; Funding acquisition, L.H.; Investigation, Y.Z. and J.L.; Methodology, L.H. and Y.P.; Software, J.L.; Validation, Q.C.; Writing—original draft, L.H.; Writing—review & editing, X.Z., W.X. and H.X.

**Funding:** The authors gratefully acknowledge the funding of the National Natural Science Foundation of China (nos. 21563011 and nos. 31672044) and the Natural Science Foundation of Jiang Xi Province of China (20181BAB204012), Project of Science and Technology in Guangdong Province (2015B020230012).

**Conflicts of Interest:** The authors declare no conflict of interest. The founding sponsors had no role in the design of the study; in the collection, analyses, or interpretation of data; in the writing of the manuscript; or in the decision to publish the results.

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
