# Peer review of "Exploration of Light-Controlled Chemical Behavior and Mechanism in a Macrocyclic Copper Complex Catalyst–Acetone–Glucose–Bromate–Sulfuric Acid Oscillation System"

_catalysts, doi:10.3390/catal9010065_

Round 1
Reviewer 1 Report
As continuation of the earlier studies Authors presented an oscillation system containing copper catalyst. The paper aimed in mechanistic studies and experiments were performed to characterize an effect of light on the concentration of components of the system.
However the mechanism presented in chapter 3 was proposed for the system containing malic acid but not glucose. There are also mistakes in eq. 2a (line 211) and in reactions A-C (lines 222-226). Therefore it is not clear what was in fact studied and which new aspects of the reaction mechanism were discovered. Presentation should be improved and the novel aspects of the paper should be clearly pointed out.
In my opinion in this version the paper is not suitable for publication.
Author Response
Dear reviewer,
Thank you very much for your review, your suggestion is very professional and constructive, we are very sorry for the mistakes in the paper. Please allow me to explain myself. We have studied the chemical oscillations of the two systems. In this paper, we studied Glucose-containing system, and in the other one, we studied the system containing malic acid. Due to our mistake in editing the manuscript, we mixed up the electronic documents of the two articles, which resulted in unclear expressions. We are terribly sorry for our carelessness and we have corrected this. Errors in lines 211 and 222-226 of the chemical equation have been corrected in the new manuscript. We apologize for our mistake again. Your suggestion is very helpful to us, they have important guiding significance for the writing of this paper and to improve the quality of its publication.
Thank you again for your advice and your valuable time, hoping to learn more from you.
Kind regards,
Qujin Cui

Reviewer 2 Report
The authors have shown a very interesting work to illustrate the effect of UV light on the glucose (Glu)-acetone (Act)-sodium bromate (NaBrO3)-sulfuric acid (H2SO4) oscillation system. The experimental results and the mathematical simulations clearly show a decrease in the amplitude of the chemical oscillation under UV illumination which is attributed to the increasing concentration of bromide ions in the system. This is a very solid paper and contributes to the research area devoted to the study of the effect of light interference on the oscillatory reaction. The manuscript needs a round of English editing before publication.
Author Response
Dear reviewer,
Thank you very much for your review, it is a great honor that our manuscript has received your recognition and support. Your suggestion is very professional and constructive, we are very sorry for the problem of English editing and have asked a professional organization to modify it. Now,all grammatical problems were corrected and the paper work has been finished. Your suggestion is very helpful to us, they have important guiding significance for the writing of this paper and to improve the quality of its publication.
Thank you again for your advice and your valuable time, hoping to learn more from you.
Kind regards,
Qujin Cui

Reviewer 3 Report
This paper (catalysts-408256) describes the mechanism of light-controlled BZ reaction catalyzed by macrocyclic copper complex. BZ reaction is an attractive research topic from basic chemistry. In this paper, authors discuss effect of UV irradiation to BZ reaction solution and claim photochemical bond cleavage of Br2 to Br radicals is the cause. However, I wonder if this paper is suitable for article of Catalyst because this paper does not focus on the cooper catalyst. Moreover, authors should add some experiments to the manuscript in order to come to their conclusion. Therefore, I recommend acceptance of this manuscript as an article of Catalyst after major revision if editor of Catalyst recognize the subject of this paper is suitable for Catalysts.
1) Authors should indicate dependence of wavelength of irradiation to the reaction solution. Wavelength range to influence on the reaction would be consistent with absorption wavelength of Br2.
2) Authors should elucidate the influence of light irradiation to reaction is not caused by disturbance of electric double-layer. At least, procedure of light irradiation to the electrode should be described in experimental section.
3) In figure 3 of page 3, when light irradiation was stopped, reduction potential was not changed. Authors should explain the reason of that.
4) In this paper, there are many mistakes in equations and figures.
Page8, line 211
In eq. 2a, [CuL]2+ is missing in right side.
Page8, line 224
Cu2+ is missing in left side.
In many equations in page 8
“Light”, “k” are overlapped on allows.
Page 2
Excited state of Ru(bpy)3 should be expressed by [Ru(bpy)3]3+*.
Page 9, Figure 6
A and B are missing.
Author Response
Dear reviewer,
Thank you very much for your review, your suggestions are very professional and constructive, they have important guiding significance for the writing of this paper and to improve the quality of its publication. We are very sorry for our negligence. We have studied comments carefully and have made correction which we hope meet with your approval. Here will answer your comments one by one:
Point 1: “Authors should indicate dependence of wavelength of irradiation to the reaction solution. Wavelength range to influence on the reaction would be consistent with absorption wavelength of Br2.”
Response 1: The bond energy of Br-Br is 193.9kJ/mol, and the absorption wavelength of Br2 is 630nm through calculation. The emission wavelength of ultraviolet light source used in this experiment is 254nm and 365nm, which is close to the absorption wavelength of Br2 and can have an effect on the dissociation of Br2, so it will have an impact on the concentration of Br- in the reaction process.
Point 2: “Authors should elucidate the influence of light irradiation to reaction is not caused by disturbance of electric double-layer. At least, procedure of light irradiation to the electrode should be described in experimental section.”
Response 2: Two indicator electrodes, platinum electrode and bromide ion electrode, were used in the experiment. The two experimental results all showed that ultraviolet light would affect the concentration of Br-, and the bromide ion electrode was a concentration difference electrode, and there was no double electric layer. Therefore, it could be explained that the effect of light on the reaction was not caused by the double electric layer.
Point 3: “In figure 3 of page 3, when light irradiation was stopped, reduction potential was not changed. Authors should explain the reason of that.”
Response 3: The oscillating system of BZ is very sensitive, and any slight influence will cause potential changes. At present, the process of reaction cannot be controlled, nor can the final resting position of the point be controlled, so it is normal that the reduction potential does not change.
Point 4: “In this paper, there are many mistakes in equations and figures.”
Response 4: The errors in the equations on page 8, line 211, line 224 and page 2, and the problems in figure 6 on page 9 have been corrected in the manuscript.
We look forward to hearing from you regarding our submission,we would glad to respond to any further questions and comments that you may have. Thank you again for your advice and your valuable time, your suggestion is very helpful to us, hoping to learn more from you.
Kind regards,
Qujin Cui

Round 2
Reviewer 1 Report
Authors corrected mistakes and improved presentation of the manuscript. The obtained results are original and can be published in present form.